# Molecular Classification and Clinical Outcomes in Endometrial Cancer: Real-World Evidence from a Tertiary Care Center

**DOI:** 10.3390/cancers18020181

**Published:** 2026-01-06

**Authors:** Tanadon Salakphet, Prapaporn Suprasert, Tip Pongsuvareeyakul, Chinachote Teerapakpinyo, Surapan Khunamornpong

**Affiliations:** 1Division of Gynecologic Oncology, Department of Obstetrics and Gynecology, Faculty of Medicine, Chiang Mai University, Chiang Mai 50200, Thailand; tanadon.s@cmu.ac.th; 2Division of Gynecologic Pathology, Department of Pathology, Faculty of Medicine, Chiang Mai University, Chiang Mai 50200, Thailand; tang_tip@hotmail.com (T.P.); skhunamo@yahoo.com (S.K.); 3Chula GenePRO Center, Faculty of Medicine, Chulalongkorn University, Bangkok 10330, Thailand; chinachote.t@chula.ac.th

**Keywords:** endometrial carcinoma, molecular classification, mismatch repair, p53, POLE mutation, survival, Thailand, real-world evidence

## Abstract

This study evaluated 184 Thai patients with endometrial carcinoma and available molecular data to assess its prognostic relevance in real-world practice. Tumors were classified into POLE-mutated, mismatch repair–deficient (dMMR), p53-abnormal (p53-abn), and no specific molecular profile (NSMP) groups. Survival outcomes differed by subtype: POLE-mutated tumors showed an excellent prognosis, while p53-abnormal tumors had the poorest outcomes. Even with selective testing, molecular classification effectively predicted prognosis. These findings support the routine use of MMR and p53 immunohistochemistry, with targeted POLE sequencing for intermediate- and high-risk cases, to guide personalized management in resource-limited settings.

## 1. Introduction

Endometrial cancer (EC) is the fourth most common malignancy among women worldwide, with the highest incidence reported in North America (ASR 22.3 per 100,000) [1]. In Asia, the incidence is lower (ASR 5.7 per 100,000), while in Thailand, it is slightly higher (ASR 6.9 per 100,000) and has shown a steady increase over the past decade [1]. Although the incidence in Thailand remains lower than in Western countries, the rising trend underscores its growing public health importance and highlights the need for a better understanding of its clinical and molecular characteristics.

A major advance in EC management followed the publication of The Cancer Genome Atlas (TCGA) molecular classification, which identified four prognostic subtypes: POLE ultramutated, microsatellite instability (MSI) hypermutated, copy-number low, and copy-number high [2]. This system provided more accurate prognostic predictions and helped guide adjuvant treatment decisions. Because comprehensive TCGA profiling is costly and technically demanding, simplified classifiers such as ProMisE (Proactive Molecular Risk Classifier for Endometrial Cancer) [3] and TransPORTEC [4] were developed. These approaches combine immunohistochemistry (IHC) for MMR proteins and p53 with targeted POLE sequencing, allowing tumors to be assigned to the same four molecular groups: POLE ultramutated, MMR-deficient (dMMR), p53-abnormal (p53abn), and no specific molecular profile (NSMP).

In recognition of its clinical utility, the WHO 2020 classification recommended incorporating molecular subtyping into routine pathological reporting [5], and the FIGO 2023 revision integrated molecular data into staging to enhance prognostic accuracy [6]. The ESGO–ESTRO–ESP 2021 guidelines also emphasize molecular classification in risk assessment and adjuvant planning, recommending POLE testing for intermediate or high intermediate risk cases, as POLE-ultramutated tumors exhibit excellent outcomes and may justify treatment de-escalation [7].

However, evidence from real-world Asian populations remains limited, particularly in Thailand, where incidence rates are increasing. Therefore, this study aimed to evaluate clinical outcomes and clinicopathologic correlations by molecular subtype among endometrial cancer patients treated at a tertiary care center in Thailand, providing region-specific real-world evidence to support the integration of molecular classification into routine clinical practice.

## 2. Materials and Methods

### 2.1. Study Design and Patient Population

This study was designed as a retrospective observational cohort reflecting real-world clinical practice. It was approved by the Institutional Ethics Committee of Chiang Mai University. Medical records of all patients with histologically confirmed endometrial carcinoma (EC) who underwent primary surgical treatment at Chiang Mai University Hospital between January 2015 and December 2023 were retrospectively reviewed. Given the retrospective nature of the study, analyses were exploratory and aimed to describe clinicopathologic characteristics and prognostic patterns rather than to establish causal relationships.

Most patients were diagnosed preoperatively by fractional curettage or endometrial sampling, with diagnostic hysteroscopy performed selectively based on clinical indication, resource availability, and operator expertise during the study period. Final histopathologic classification and FIGO staging were determined from definitive hysterectomy specimens.

Standard surgical management consisted of total hysterectomy with bilateral salpingo- oophorectomy and peritoneal washing. Pelvic and/or para-aortic lymph node assessment (sampling or lymphadenectomy) was performed according to tumor risk factors, intraoperative findings, patient comorbidities, and prevailing international guidelines at the time of treatment, reflecting risk-adapted real-world practice over the extended study period. All histopathologic diagnoses were verified by two gynecologic pathologists (SK, TP) to ensure consistency.

Patients were eligible if they had histologically confirmed endometrial carcinoma with available immunohistochemistry (IHC) results for mismatch repair (MMR) proteins and/or p53. For patients classified as intermediate- or high–intermediate-risk according to the ESGO–ESTRO–ESP 2021 guidelines [7] who underwent surgery between January 2022 and December 2023, archival formalin-fixed paraffin-embedded (FFPE) tissue blocks were retrieved for POLE mutation analysis. POLE testing was restricted to this period to ensure adequate DNA quality, as older FFPE specimens frequently demonstrated DNA degradation that could compromise the reliability of exonuclease domain sequencing. POLE mutation analysis was performed and interpreted by an expert molecular geneticist (CT). Patients were excluded if histologic slides or FFPE tissue blocks were unavailable or unsuitable for molecular testing, or if post-treatment clinical outcome data were incomplete.

### 2.2. Testing for Molecular Classification

#### 2.2.1. Immunohistochemistry for MMR and p53

IHC staining was performed using the BenchMark ULTRA IHC/ISH platform (Ventana Medical Systems, Roche Diagnostics, Tucson, AZ, USA). MMR status was assessed using a validated two-antibody approach (PMS2 and MSH6) instead of the conventional four-marker panel (MLH1, PMS2, MSH2, MSH6), providing comparable accuracy with greater cost efficiency [8,9]. Primary antibodies against PMS2 (clone A16-4, Ventana; OptiView detection with amplification) and MSH6 (clone SP93, Ventana; OptiView detection) were applied. Loss of MMR expression was defined as the complete absence of nuclear staining in tumor cells with retained staining in internal non-neoplastic tissue; such cases were classified as dMMR.

For p53 expression, a primary monoclonal antibody (clone DO-7, DAKO, dilution 1:100) was used. Staining results were categorized as wild-type or abnormal. Wild-type p53 expression showed variable nuclear staining intensity among tumor cells. Abnormal expression was defined by one of three patterns: (1) diffuse strong nuclear positivity in ≥80% of tumor cells (over-expression), (2) complete absence of nuclear staining (null pattern), or (3) cytoplasmic accumulation of p53 with or without variable nuclear staining. Any of these abnormal patterns was interpreted as p53abn [10].

#### 2.2.2. POLE Mutation Analysis

Histopathological evaluation was performed by board-certified pathologists to ensure adequate tumor content in all FFPE specimens. Genomic DNA was extracted using the cobas^®^ DNA Sample Preparation Kit (Roche Diagnostics, Mannheim, Germany). POLE mutation analysis was conducted on 500–2000 ng of extracted DNA using the Idylla™ POLE–POLD1 Mutation Assay (Biocartis, Mechelen, Belgium), a fully automated, real-time PCR system that detects 17 hotspot mutations within the POLE exonuclease domain (exons 9, 11, 13, and 14), encompassing the majority of clinically relevant variants [11,12].

Samples that tested positive for a POLE mutation or yielded invalid Idylla results were subjected to confirmatory bidirectional Sanger sequencing according to standard laboratory protocols. Sequence variants were reported according to Human Genome Variation Society (HGVS) nomenclature, based on the reference sequence NM_006231.2 [13]. All molecular analyses were performed and interpreted by an expert in gene analysis (CT).

#### 2.2.3. Final Molecular Subtype Assignment

Final molecular subtype assignment was determined using a hierarchical algorithm (POLE → MMR → p53 → NSMP), adapted to reflect the incomplete molecular testing performed in routine practice. Tumors with pathogenic POLE mutations confirmed by Sanger sequencing were classified as POLE-mutated. Cases showing loss of one or more MMR proteins were categorized as dMMR, regardless of p53 status, when POLE testing was negative or unavailable. Tumors with abnormal p53 staining (overexpression, null, or cytoplasmic) were assigned to the p53-abn group when MMR was intact or when MMR testing had not been performed, consistent with selective testing during the study period. NSMP was reserved for tumors with documented pMMR, wild-type p53, and negative POLE sequencing. This pragmatic approach ensured appropriate subtype assignment despite real-world variability in test completeness.

### 2.3. Treatment, Follow-Up, and Outcome Definitions

Adjuvant therapy was administered according to multidisciplinary recommendations from the gynecologic oncology team, taking into account clinical risk factors and patient preferences. Following completion of primary treatment, patients entered a structured surveillance program consisting of follow-up visits every three months in the first year, every four months in the second year, every six months during years three to five, and annually thereafter. Each visit included detailed history taking, physical and pelvic examinations, and imaging studies such as computed tomography (CT) when clinically indicated.

Progression-free survival (PFS) was defined as the time from surgery, or from initiation of neoadjuvant chemotherapy, to the date of radiologically or pathologically confirmed recurrence or last follow-up. Overall survival (OS) was measured from the same starting point to death from any cause or last documented contact. Vital status was verified using the national population registry. Baseline clinicopathologic characteristics and survival outcomes were systematically recorded for statistical analysis.

### 2.4. Sample-Size Calculation

The sample size was estimated a priori to ensure adequate power to detect survival differences among the four molecular subtypes (POLE-ultramutated, dMMR, p53-abn, and NSMP). Based on the log-rank test for equality of survival curves (two-sided α = 0.05, 80% power) and using the Freedman formula [14], the required total sample size was approximately 180 patients. The calculation assumed a hazard ratio of 2.5 between the most and least favorable groups and an event rate of 20%, parameters derived from previous large-scale molecular studies validating the TCGA classification [12,15]. Detailed assumptions and computational steps are provided in Appendix B.

### 2.5. Statistical Analysis

All statistical analyses were performed using IBM SPSS Statistics version 23.0 (IBM Corp., Armonk, NY, USA). Descriptive statistics were used to summarize patient demographics and clinicopathologic characteristics. Continuous variables are presented as mean ± standard deviation, and categorical variables as frequencies and percentages. Comparisons between categorical variables were conducted using the chi-square test or Fisher’s exact test, as appropriate. Variables demonstrating a *p* value < 0.05 in univariate analysis were subsequently entered into multivariable binary logistic regression models to identify independent factors associated with MMR deficiency and p53 abnormality. Survival outcomes were estimated using the Kaplan–Meier method and compared across molecular subtypes using the log-rank test. All statistical tests were two-sided, and a *p*-value < 0.05 was considered statistically significant.

## 3. Results

### 3.1. Clinicopathologic Characteristics

Between January 2015 and December 2023, a total of 803 patients were diagnosed with endometrial carcinoma at Chiang Mai University Hospital; 184 patients met the inclusion criteria and were included in the final analysis (Appendix A). The mean age was 61.3 ± 9.7 years, and the mean body mass index (BMI) was 24.5 ± 5.1 kg/m^2^. The most common presenting symptom was postmenopausal bleeding (82.6%).

Comorbidities were present in 70.1% of patients, most frequently diabetes mellitus, hypertension, and dyslipidemia, and 15.2% had a history of another malignancy, most commonly breast cancer. Preoperative tissue diagnosis was obtained by fractional curettage or endometrial sampling in 92.9% of cases.

Most patients underwent exploratory laparotomy (89.7%), while 10.3% were managed laparoscopically. Pelvic lymph node assessment was performed in 75.0%, and paraaortic lymph node assessment in 26.6%. Complete macroscopic resection was achieved in 91.8% of patients.

According to FIGO 2018 staging, disease stages were distributed as follows: IA 32.6%, IB 25.5%, II 6.0%, IIIA 8.7%, IIIB 3.8%, IIIC1 8.7%, IIIC2 5.4%, and IVB 9.2%. Endometrioid carcinoma was the predominant histologic subtype (57.1%), followed by high-grade serous carcinoma (30.4%), clear cell carcinoma (6.5%), and carcinosarcoma (6.0%). High-grade tumors (grade 3) accounted for 63.0% of cases.

Myometrial invasion ≥ 50% was observed in 38.6%, lymphovascular space invasion (LVSI) was present in 59.2%, and endocervical involvement occurred in 23.4%. Pelvic and paraaortic lymph node metastases were identified in 14.7% and 3.8%, respectively. Based on ESGO–ESTRO–ESP 2021 risk classification [7], 57.6% of patients were categorized as high-risk.

Neoadjuvant chemotherapy was administered in 7.6%, and 79.3% received adjuvant treatment following surgery. During follow-up, 15.8% experienced disease recurrence, and 28.3% died.

### 3.2. Molecular Classification and Correlation with Clinicopathologic Features

The immunohistochemistry (IHC) results for mismatch repair (MMR) proteins and p53, together with POLE mutation testing, which collectively formed the basis of final molecular subtype assignment, are summarized in Table 1. Among the 184 patients, molecular testing combinations included MMR alone in 13.0%, p53 alone in 29.9%, POLE alone in 13.6%, MMR plus p53 in 38.0%, MMR plus POLE in 1.1%, p53 plus POLE in 1.6%, and combined MMR, p53, and POLE testing in 2.7%. Overall, 38.6% of tumors were classified as MMR-deficient, 45.0% showed abnormal p53 expression—most commonly overexpression (35.9%), followed by null (8.2%) and cytoplasmic (1.1%) patterns—and POLE mutations were identified in 2.2% of cases. Three patients (1.6%) were classified as NSMP (pMMR + p53 wild type + POLE negative), all presenting with stage I endometrioid carcinoma. Each received adjuvant therapy and remained disease-free, with progression-free survival (PFS) of 1.77–2.02 years (Appendix A).

Among 35 patients tested for POLE mutation, six were initially positive, and two were invalid by the Idylla™ POLE–POLD1 Mutation Assay. Subsequent confirmatory Sanger sequencing verified four POLE-mutated cases (2.2%) (Appendix A). All four patients (aged 56–75 years) had endometrioid carcinoma, one grade 1, one grade 2, and two grade 3, with deep myometrial invasion in three cases. Two received external beam radiation therapy (EBRT) + vaginal brachytherapy (VBT), and two received VBT alone; all remained alive and recurrence-free at last follow-up.

Immunohistochemistry (IHC) results included mismatch repair (MMR) protein status (deficient vs. proficient) and p53 expression patterns (wild-type, overexpression, null, or cytoplasmic), together with POLE mutation status when available. These molecular findings were integrated to assign final molecular subtypes according to a predefined hierarchical algorithm. All IHC slides were independently reviewed by two gynecologic pathologists.

### 3.3. Survival Outcomes by Molecular Subtype

Survival analysis stratified by molecular subtype (Figure 1a,b) demonstrated distinct differences in outcomes. POLE-mutated tumors showed no recurrence or death during follow-up. dMMR tumors exhibited intermediate progression-free and overall survival, whereas p53-abnormal (p53-abn) tumors had the poorest outcomes. Overall survival differed significantly among molecular subtypes (Figure 1b), driven primarily by inferior survival in the p53-abn group. In contrast, differences in progression-free survival did not reach statistical significance (Figure 1a). All three patients with NSMP remained recurrence-free throughout the observation period.

### 3.4. Clinicopathologic Correlates of MMR and p53 Status

Associations between molecular markers and clinicopathologic features are summarized in Table 2. Among 184 patients, dMMR tumors were predominantly endometrioid (86.7%) compared with 13.3% in the pMMR group (*p* < 0.001), whereas non-endometrioid histology was more frequent in pMMR tumors (65.5% vs. 26.9%). dMMR tumors were also associated with older age (>60 years, *p* = 0.006) and low &intermediate risk (*p* < 0.001). In multivariate analysis, histologic type remained the only independent predictor of dMMR (adjusted OR = 15.215, 95% CI 4.992–46.374, *p* < 0.001).

For p53 status, p53-abn was more common in older patients (>60 years, *p* < 0.001), comorbidities (*p* < 0.001), in non-endometrioid histology (*p* < 0.001), in high-grade tumors (*p* < 0.001), and among patients receiving adjuvant radiation (*p* < 0.001) and adjuvant chemotherapy (*p* = 0.004). Histologic type again emerged as the sole independent predictor (adjusted OR = 79.416, 95% CI 10.599–595.046, *p* < 0.001).

Other parameters, including BMI, residual disease, depth of myometrial invasion, LVSI, peritoneal cytology, FIGO stage, risk group, and adjuvant therapy, showed no significant correlation with either MMR or p53 status.

## 4. Discussion

### 4.1. Distribution of Molecular Subtypes

This study provides real-world evidence on the molecular classification of EC among Thai patients, integrating MMR, p53, and POLE testing. Although molecular profiling is now central to EC prognostication, its routine use remains limited in many low- and middle-income settings because of cost and infrastructure barriers. In our cohort of 184 surgically treated patients, roughly one-quarter of all EC cases diagnosed during the study period underwent molecular testing, reflecting the gradual adoption of molecular diagnostics into standard workflows.

The lower frequency of POLE mutations observed in our cohort compared with Western series (typically 5–10%) [2,12,16] likely reflects selective rather than universal testing, as POLE sequencing was restricted to patients with intermediate- or high–intermediate-risk disease according to ESGO–ESTRO–ESP criteria [7]. Consequently, the true prevalence of POLE-mutated EC in our population may be underestimated. The proportion of dMMR tumors in our cohort (38.6%) was comparable to that reported in the PORTEC-3 molecular analysis (34–36%) [15]. In contrast, p53-abnormal (p53-abn) tumors were more prevalent than the 23% reported in PORTEC-3 [15]. This difference likely reflects referral bias toward patients with high-grade disease at our tertiary center, as well as the selective use of p53 immunohistochemistry in clinically aggressive tumors, such as high-grade serous carcinoma.

### 4.2. Survival Outcomes by Molecular Subtype

Survival outcomes differed across the final molecular subtypes, showing patterns consistent with established literature. POLE-mutated tumors demonstrated the most favorable prognosis, with no recurrences or deaths, in agreement with outcomes from the TCGA and León-Castillo et al. cohorts [2,12]. dMMR tumors showed intermediate outcomes, reflecting the relatively favorable behavior of hypermutated, immune-reactive carcinomas. In contrast, p53-abn tumors exhibited the poorest prognosis. Importantly, overall survival differed significantly across molecular subtypes, driven primarily by the markedly inferior OS observed in the p53-abn group, consistent with the aggressive nature of the copy-number–high subtype reported in PORTEC-3 and guideline-based analyses [7,15].

In contrast, progression-free survival differences did not reach statistical significance, likely reflecting limited statistical power due to the small number of patients in certain molecular subgroups, particularly POLE-mutated and NSMP tumors. Accordingly, these findings should be interpreted as reflecting overall prognostic trends across the major molecular categories, rather than definitive subgroup-specific comparisons. Nonetheless, the Kaplan–Meier survival patterns demonstrated clinically meaningful separation that aligns with established molecular prognostic hierarchies.

### 4.3. Methodological Considerations for POLE Testing

POLE mutation analysis in this study employed the Idylla™ POLE–POLD1 Mutation Assay (Biocartis) as a first-line screening tool, with bidirectional Sanger sequencing performed to confirm positive or invalid results. This two-step workflow was selected for practicality, as the Idylla platform provides a fully automated, cartridge-based process that requires minimal technical expertise and yields results within approximately two hours [11,17]. The assay targets 17 hotspot mutations within exons 9, 11, 13, and 14 of the POLE exonuclease domain, encompassing most clinically relevant variants but excluding rare or novel mutations outside these regions [11,12].

The low POLE mutation frequency observed (2.2%) likely reflects selective testing criteria rather than assay limitations, since molecular analysis was restricted to intermediate- and high-risk cases. The combination of Idylla screening with confirmatory Sanger sequencing represents a reliable and cost-effective strategy, with previous studies reporting >95% concordance with next-generation sequencing for known hotspot variants [11,18].

All four POLE-mutated tumors in our cohort were endometrioid carcinomas, spanning grades 1–3 and stages IA–IB, with deep myometrial invasion in most cases. None recurred after standard adjuvant therapy, consistent with the indolent clinical course characteristic of POLE-ultramutated ECs, even when high-grade histology is present [12,13]. These findings support ongoing efforts to de-escalate adjuvant therapy in this subgroup [4], though larger studies with broader molecular testing are needed to define prevalence and prognostic implications in Asian populations.

### 4.4. Clinicopathologic Correlates of MMR and p53 Status

Our results reaffirm the strong association between histologic type and molecular phenotype in endometrial carcinoma. dMMR tumors were predominantly endometrioid, occurred more frequently in older women, and were commonly classified as low- or intermediate-risk, consistent with the hypermutated, microsatellite-unstable subtype described in prior studies [19,20]. In a U.S. cohort of 223 dMMR endometrial cancers, Chase et al. [19] reported a mean patient age of 64 years and endometrioid histology in more than 90% of cases, closely mirroring our findings. Similarly, Jumaah et al. [20] demonstrated that dMMR tumors are strongly associated with older age, endometrioid morphology, and lower tumor grade, supporting a distinct pathogenetic pathway within this molecular subgroup.

In contrast, p53-abnormal tumors in our cohort were predominantly non-endometrioid, particularly serous and clear cell carcinomas, and were largely high grade, consistent with their classification within the copy-number–high molecular subtype [2,15]. Recent data from Vaziri-Fard et al. [21] further corroborate these observations, showing that p53-abnormal endometrioid carcinomas display biological aggressiveness comparable to that of serous carcinoma.

In our multivariate logistic regression, histologic type was the only independent predictor for both MMR deficiency and p53 abnormality, underscoring the value of morphology as a surrogate indicator when molecular testing is incomplete. Other parameters, including BMI, comorbidities, residual disease, myometrial invasion, LVSI, peritoneal cytology, stage, ESGO–ESTRO–ESP risk group, and adjuvant therapy, showed no independent association with molecular status.

These findings differ from a recent meta-analysis of 4776 EC cases, which reported significant associations between dMMR and higher tumor grade and LVSI, and between the p53-abn (copy-number–high) subtype and advanced stage, >50% myometrial invasion, increased LVSI, and lymph-node metastasis [22]. The lack of similar associations in our study likely reflects selective molecular testing and limited statistical power due to smaller subgroup sizes.

### 4.5. Strengths and Limitations

The strengths of this study lie in its integration of real-world molecular testing with comprehensive clinicopathologic correlation and validated survival outcomes. All histologic slides were reviewed by gynecologic pathologists, and POLE sequencing results were confirmed by bidirectional Sanger analysis, ensuring high diagnostic accuracy. Moreover, survival data were verified through the national civil registry, minimizing loss to follow-up.

However, several limitations should be acknowledged. First, the retrospective design inherently limits causal inference and is subject to information and selection bias. Second, molecular testing was not uniformly performed across the entire study period, reflecting real-world practice during a transitional phase in the adoption of molecular classification. In particular, POLE mutation analysis was restricted to more recent surgical cases because adequate DNA quality could not be reliably obtained from older formalin-fixed, paraffin-embedded (FFPE) tissue blocks. This constraint likely resulted in an underestimation of the true prevalence of POLE-mutated tumors in earlier cases. Third, surgical staging practices varied over time, in accordance with evolving international guidelines and individualized, risk-adapted clinical decision-making, which may have introduced heterogeneity in clinicopathologic assessment. Fourth, the small number of patients in certain molecular subgroups—especially POLE-mutated and NSMP tumors—limited statistical power for subgroup analyses and warrants cautious interpretation of subgroup-specific outcomes. Finally, the relatively short median follow-up duration in the POLE-mutated and NSMP groups may have led to underestimation of late recurrences.

## 5. Conclusions

Despite these limitations, the findings underscore the prognostic value of integrated molecular classification in endometrial carcinoma, even when applied selectively. MMR and p53 immunohistochemistry remain practical, low-cost frontline assays for molecular stratification, while POLE sequencing can be reserved for intermediate- and high-risk endometrioid cases, consistent with WHO 2020 and ESGO–ESTRO–ESP 2021 recommendations [5,7]. Future multicenter studies in Southeast Asia should aim to define regional molecular subtype distributions, establish cost-effective testing algorithms, and evaluate molecularly guided adjuvant strategies to optimize outcomes in resource-limited healthcare settings.

## Figures and Tables

**Figure 1 cancers-18-00181-f001:**
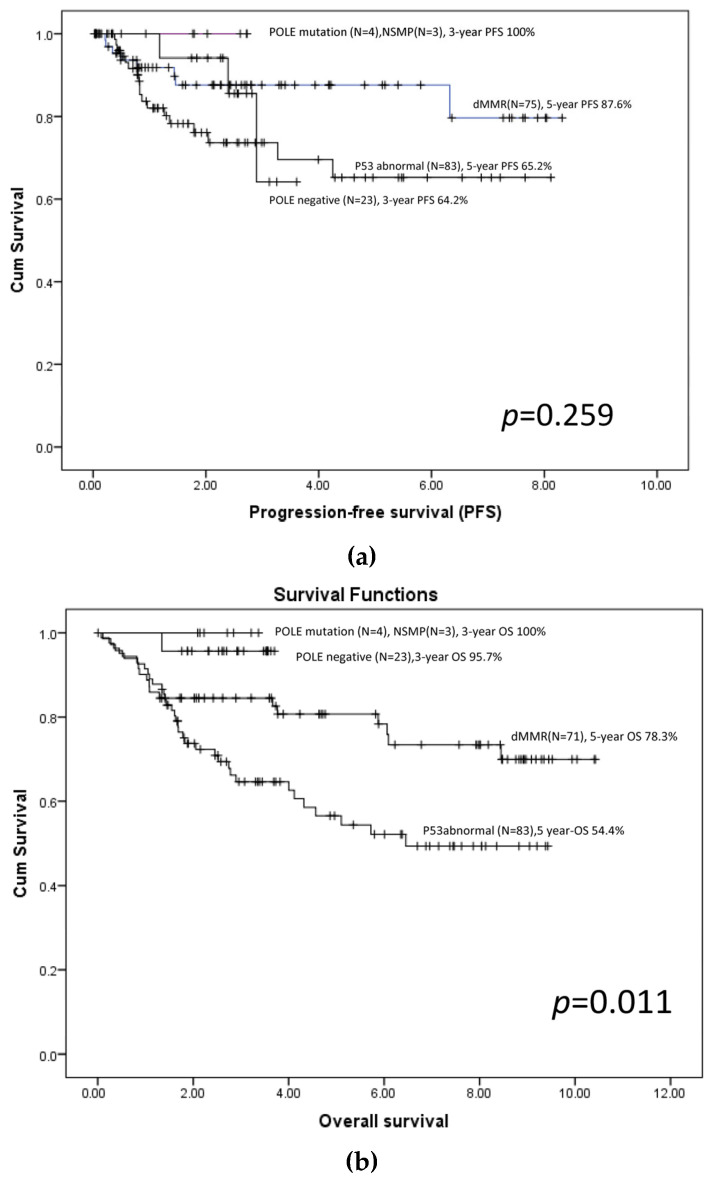
Progression-free survival (PFS) (**a**) and overall survival (OS) (**b**) by final molecular subtype. (**a**) Kaplan–Meier curves for progression-free survival (PFS) among POLE-mutated, POLE-negative, mismatch repair–deficient (dMMR), non-specific molecular profile (NSMP) and p53-abnormal subtypes. PFS differences were not significant (log-rank *p* = 0.259), with no progression in the POLE-mutated and NSMP groups and the lowest PFS in the p53-abnormal group. (**b**) Overall survival (OS) differed significantly across subtypes (log-rank *p* = 0.011). POLE-mutated tumors showed 100% 3-year OS, POLE-negative tumors had favorable survival, dMMR had intermediate outcomes, and p53-abnormal tumors had the poorest OS.

**Table 1 cancers-18-00181-t001:** Immunohistochemistry Results and Final Molecular Subtype Assignment in 184 Patients with Endometrial Carcinoma.

Data	Number (%)
Testing coverage	
MMR	24 (13.0)
P53	55 (29.9)
POLE	25 (13.6)
MMR + p53	70 (38.0)
MMR + POLE	2 (1.1)
P53 + POLE	3 (1.6)
MMR + p53 + POLE	5 (2.7)
IHC and molecular findings	
dMMR	24 (13)
Abnormal p53	55 (29.9)
POLE negative	22 (12.0)
POLE mutation	2 (1.1)
POLE invalid	1 (0.5)
dMMR + wild type P53	45 (24.5)
dMMR + POLE negative	2 (1.1)
dMMR + wildtype P53 + POLE mutation	1 (0.5)
pMMR + P 53 abnormal	26 (14.1)
pMMR+ wild type p53 + POLE negative (NSMP)	3 (1.6)
wide type P53 + POLE mutation	1 (0.5)
P53 abnormal + POLE negative	2 (1.1)
Final Molecular Subtype Assignment	
Pole mutation	4 (2.2)
Pole negative	23 (12.5)
p53 abnormal	83 (45.1)
dMMR	71 (38.6)
NSMP	3 (1.6)
Summary of results by individual molecular test	
MMR	
Not done	83 (45.1)
pMMR	29 (15.8)
dMMR	72 (39.1)
P53	
Not done	51 (27.7)
Abnormal *	85 (46.2)
Wild type	50 (30.3)
POLE	
Not done	149 (81.0)
Done	35 (19.0)
Not detected	29 (15.8)
Detected	4 (2.1)
Invalid	2 (1.1)

* 35.9% overexpression, 8.2% null, 1.1% cytoplasmic. MMR, mismatch repair gene; POLE, DNA polymerase epsilon gene; dMMR, deficient mismatch repair; pMMR, proficient mismatch repair; NSMP, non specific molecular profile.

**Table 2 cancers-18-00181-t002:** Univariate and Multivariate Logistic Regression Analysis of Clinicopathological Factors Associated with MMR and p53 Status.

Factors	MMR							p53						
	dMMR (%)	pMMR (%)	Total	Unadjusted OR * (95%CI)	*p* Value	Adjusted OR ** (95%CI)	*p* Value	Wild	Abnormal	Total	Unadjusted OR * (95%CI)	*p* Value	Adjusted OR ** (95%CI)	*p* Value
Age (years)														
≤60 years	52 (81.3)	12 (18.8)	64 (63.4)	3.683 (1.496–9.070)	0.006	2.611 (0.888–7.677)	0.081	33 (61.1)	21 (38.9)	54 (40.6)	5.731 (2.663–12.332)	<0.001	7.952 (1.208–47.697)	0.031
>60 years	20 (54.1)	17 (45.9)	37 (36.6)					17 (21.5)	62 (78.5)	79 (59.4)				
BMI														
<25	47 (75.8)	15 (24.2)	62 (61.4)	1.755 (0.731–4.210)	0.260	-	-	32 (43.8)	41 (56.2)	73 (54.9)	1.821 (0.886–3.742)	0.109	-	-
≥25	25 (64.1)	14 (35.9)	39 (38.6)					18 (30.0)	42 (70.0)	60 (45.1)				
UD														
None	28 (82.4)	6 (17.6)	34 (33.7)	2.439 (0.883–6.736)	0.104	-	-	25 (67.6)	12 (32.4)		5.917 (2.519–13.509)	<0.001	5.618 (0.870–36.278)	0.070
Present	44 (65.7)	23 (34.3)	67 (66.3)					25 (26.0)	71 (74.0)					
Residual disease														
None	65 (70.7)	27 (29.3)	92 (91.1)	0.688 (0.134–2.526)	0.727	-	-	45 (37.5)	75 (62.5)	120 (90.2)	0.960 (0.269–3.114)	1.000	-	-
Present	7 (77.8)	2 (22.2)	9 (8.9)					5 (38.5)	8 (61.5)	13 (9.8)				
Histology														
Endometrioid	65 (86.7)	10 (13.3)	75 (74.3)	17.463 (5.915–52.622)	<0.001	15.215 (4.992–46.374)	<0.001	46 (80.7)	11 (19.3)	57 (42.9)	75.273 (22.611–250.586	<0.001	79.416 (10.599–595.049)	<0.001
Non-endometrioid	7 (26.9)	19 (65.5)	26 (25.7)					4 (5.3)	72 (94.7)	76 (57.1)				
MI														
Less than 50%	39 (76.5)	12 (23.5)	51 (50.5)	1.674 (0.700–4.006)	0.277	-	-				1.266 (0.625–2.563)	0.590	-	-
≥50%	33 (66.0)	17 (28.7)	50 (49.5)					26 (35.1)	48 (64.9)	74 (55.6)				
LVSI														-
None	45 (73.8)	16 (26.2)	61 (60.4)	1.354 (0.565–3.244)	0.509	-	-				0.627 (0.309–1.275)	0.210	-	
Present	27 (67.5)	13 (32.5)	40 (39.6)					25 (50.0)	32 (56.1)	57 (42.9)				
Peritoneal cytology														
Negative	70 (72.9)	26 (27.1)	96 (95.0)	4.038 (0.638–25.555)	0.141	-	-	48 (40.7)	70 (59.3)	118 (88.7)	4.457 (0.962–20.653)	0.050	-	-
Positive	2 (40.0)	3 (60.0)	5 (5.0)					2 (13.3)	13 (86.7)	15 (11.3)				
Stage ***														
I&II	47 (78.3)	13 (21.7)	60 (59.4)	2.314 (0.962–5.568)	0.074	-	-	33 (44.0)	42 (56.0)	75 (56.4)	1.895 (0.917–3.918)	0.105	-	-
III&IV	25 (61.0)	16 (39.0)	41 (40.6)					17 (29.3)	41 (70.7)	58 (43.6)				
Risk														
Low&intermediate	44 (89.8)	5 (10.2)	49 (48.5)	7.543 (2.578–22.072)	<0.001	1.329 (0.298–5.919)	0.709	31 (81.6)	7 (18.4)	38 (28.6)	17.714 (6.769–46.357)	<0.001	1.848 (0.059–57.817)	0.727
High risk	28 (53.8)	24 (46.2)	52 (51.5)					19 (20.0)	76 (80.0)	95 (71.4)				
Adjuvant RT														
None	31 (77.5)	9 (22.5)	40 (75.5)	0.287 (0.03–2.516)	0.419	-	-	19 (32.2)	40 (67.8)	59 (84.3)	0.048 (0.006–0.398)	<0.001	0.268 (0.020–3.600)	0.320
Present	12 (92.3)	1 (7.7)	13 (24.5)					10 (90.9)	1 (9.1)	11 (15.7)				
Adjuvant CT														
None	66 (74.2)	23 (25.8)	89 (88.1)	2.870 (0.841–9.789)	0.098	-	-	44 (44.9)	54 (55.1)	98 (73.7)	3.938 (1.500–10.337)	0.004	0.546 (0.037–8.013)	0.659
Present	6 (50.0)	6 (50.0)	12 (11.9)					6 (17.1)	29 (82.9)	35 (26.3)				

* Chi-square or Fisher’s exact test. ** Binary regression (Backward, likelihood ratio). *** Figo stage 2018. MMR, mismatch repair gene; dMMR, deficient mismatch repair; pMMR, proficient mismatch repair; OR, odds ratio; CI, confidence interval; BMI, body mass index; UD, underlying disease; MI, myometrial invasion; LVSI, lymphovascular space invasion; RT, radiation therapy; CT, chemotherapy.

## Data Availability

The data presented in this study are available on request from the corresponding author.

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
