# Peer review of "Molecular Classification and Clinical Outcomes in Endometrial Cancer: Real-World Evidence from a Tertiary Care Center"

_cancers, 2026, doi:10.3390/cancers18020181_

Round 1
Reviewer 1 Report
Comments and Suggestions for Authors
This manuscript provides real-world evidence on the prognostic significance of molecular classification in endometrial carcinoma from a Thai cohort, addressing a gap in data from Asian populations. The findings show that molecular classification can The findings show that molecular classification can effectively predict prognosis. However, several issues should be addressed to strengthen the manuscript.
- As the study focuses on Thai patients, the manuscript would benefit from a clearer and more systematic comparison with major Western studies, particularly regarding molecular subtype distribution and survival outcomes. This would help clarify potential ethnic or regional differences and enhance the global relevance of the findings.
- Some molecular subgroups have extremely small sample sizes, which limits the reliability of subgroup analyses. The conclusions drawn from these underpowered groups should be substantially tempered, with greater emphasis placed on the overall prognostic trends across the major molecular categories.
Author Response
Reviewer 1
This manuscript provides real-world evidence on the prognostic significance of molecular classification in endometrial carcinoma from a Thai cohort, addressing a gap in data from Asian populations. The findings show that molecular classification can effectively predict prognosis. However, several issues should be addressed to strengthen the manuscript.
- As the study focuses on Thai patients, the manuscript would benefit from a clearer and more systematic comparison with major Western studies, particularly regarding molecular subtype distribution and survival outcomes. This would help clarify potential ethnic or regional differences and enhance the global relevance of the findings.
Response
We sincerely thank the reviewer for this valuable suggestion. In response, we have substantially revised the Discussion section to provide a clearer and more systematic comparison between our Thai real-world cohort and major Western studies.
Specifically, we now compare the distribution of molecular subtypes in our cohort with large Western datasets, including TCGA, PORTEC-3, and studies by León-Castillo et al., highlighting similarities and differences in the prevalence of dMMR, p53-abnormal, and POLE-mutated tumors. We further discuss how the lower frequency of POLE mutations and the relatively higher proportion of p53-abnormal tumors in our study may reflect selective molecular testing, referral patterns to a tertiary center, and real-world practice constraints, rather than true biological differences alone.
In addition, we expanded the discussion of survival outcomes by molecular subtype, emphasizing that although statistical significance was limited by sample size, the observed prognostic hierarchy (POLE-mutated > dMMR > p53-abnormal) closely mirrors that reported in Western trials. We also addressed potential ethnic and regional considerations, while acknowledging the need for larger multicenter Asian studies to better delineate population-specific patterns.
These revisions have strengthened the global context and clinical relevance of our findings. The corresponding changes are detailed in the revised Discussion section below:
- Distribution of Molecular Subtypes (Western vs. Thai Cohort)
Addressed in: Discussion → Distribution of Molecular Subtypes
“The lower frequency of POLE mutations observed in our cohort compared with Western series (typically 5–10%) [2,12,16] likely reflects selective rather than universal testing…” (line 389-393 in the track change version, line 328-329 in the non track change version )
“The proportion of dMMR tumors in our cohort (38.6%) was comparable to that reported in the PORTEC-3 molecular analysis (34–36%) [15].” (line 398-399 in the track change version, line 333-334 in the non track change version )
“In contrast, p53-abnormal (p53-abn) tumors were more prevalent than the 23% reported in PORTEC-3 [15]. This difference likely reflects referral bias toward patients with high-grade disease…” (line 399-400 in the track change version, line 334-335 in the non track change version )
- Survival Outcomes Compared with Western Trials
Addressed in: Discussion → Survival Outcomes by Molecular Subtype
“POLE-mutated tumors demonstrated the most favorable prognosis… in agreement with findings from TCGA and León-Castillo et al. [2,12].” (line 410-412 in the track change version, line 341-343 in the non track change version )
“Overall survival differed significantly across molecular subtypes, driven primarily by the inferior survival observed in the p53-abn group, consistent with the aggressive biology reported in PORTEC-3 and guideline-based analyses [7,15].” (line 415-418 in the track change version, line 346-349 in the non track change version )
- Clinicopathologic Correlates: Asian vs. Western Evidence
Addressed in: Discussion → Clinicopathologic Correlates of MMR and p53 Status
“Chase et al. [19] reported a mean patient age of 64 years and endometrioid histology in more than 90% of cases, closely mirroring our findings.” (line 454-455 in the track change version, line 386-387in the non track change version)
“These findings differ from a recent meta-analysis of 4,776 EC cases… [22]. The lack of similar associations in our study likely reflects selective molecular testing and limited statistical power…” (line 476-481 in track change version, line 403-406 in no track change version)
- Some molecular subgroups have extremely small sample sizes, which limits the reliability of subgroup analyses. The conclusions drawn from these underpowered groups should be substantially tempered, with greater emphasis placed on the overall prognostic trends across the major molecular categories.
Response
We thank the reviewer for this important comment and fully agree with this concern. In response, we have revised the manuscript to substantially temper interpretations derived from underpowered molecular subgroups and to emphasize overall prognostic trends across the major molecular categories.
Specifically, we revised the Discussion (Survival Outcomes by Molecular Subtype) to clarify that while overall survival differed significantly among molecular classes, progression-free survival differences did not reach statistical significance, likely due to limited statistical power in small subgroups such as POLE-mutated and NSMP tumors. We now explicitly state that subgroup-specific findings should be interpreted as descriptive trends rather than definitive conclusions, and that the primary value of our data lies in demonstrating alignment with established molecular prognostic hierarchies.
The revised manuscript now state ((line 419-425 in the track change version, line 350-356 in the non track change version) (part survival outcome by molecular in discussion section)
“In contrast, progression-free survival differences did not reach statistical significance, likely reflecting limited statistical power due to the small number of patients in certain molecular subgroups, particularly POLE-mutated and NSMP tumors. Accordingly, these findings should be interpreted as reflecting overall prognostic trends across the major molecular categories, rather than definitive subgroup-specific comparisons. Nonetheless, the Kaplan–Meier survival patterns demonstrated clinically meaningful separation that aligns with established molecular prognostic hierarchies.”
In addition, we strengthened the Limitations section to clearly acknowledge the methodological constraints highlighted by the reviewer. The revised manuscript now states: (line 508-511 in the track change version, line 426-429 in the non track change version)
“….Fourth, the small number of patients in certain molecular subgroups,especially POLE-mutated and NSMP tumors—limited statistical power for subgroup analyses and warrants cautious interpretation of subgroup-specific outcomes….”

Reviewer 2 Report
Comments and Suggestions for Authors
The study is promising, as it addresses the crucial aspect of molecular classification of endometrial cancer.
However, the M&Ms leave much to be desired.
First of all, the study is retrospective, which severely limits the validity of the results.
Then, the idea of ​​"at discretion" pelvic and/or lumboaortic lymphadenectomy/sampling doesn't sound right to me. Either you follow the guidelines or you don't.
Again. But if the study was conducted between January 2015 and December 2023, why were only tumors classified as intermediate- or high-intermediate-risk per ESGO–ESTRO–ESP 2021 guidelines who underwent surgery between 2022 and 2023, archival formalin-fixed paraffin-embedded (FFPE) tissue blocks retrieved for POLE 100
mutation testing?
The English language leaves much to be desired in many places and is unclear or approximate (see lines 188-190).
The authors state that: Most patients were diagnosed preoperatively by fractional curettage or endometrial sampling.
But what if diagnostic hysteroscopy and avoiding blind diagnosis have been discussed for years (see FIGO guidelines)?
The paragraph from lines 192 to 196 needs to be completely rewritten and clarified. We need to be precise in staging and percentages, avoiding approximations.
This work seems to me, in its premise, like an excellent restaurant dish,
and then it turns out like a frozen-food soup.
I've thoroughly revised it.
Author Response
Comments and Suggestions for Authors
Reviewer 2
The study is promising, as it addresses the crucial aspect of molecular classification of endometrial cancer.
However, the M&Ms leave much to be desired.
First of all, the study is retrospective, which severely limits the validity of the results.
Then, the idea of ​​"at discretion" pelvic and/or lumboaortic lymphadenectomy/sampling doesn't sound right to me. Either you follow the guidelines or you don't.
Again.
But if the study was conducted between January 2015 and December 2023, why were only tumors classified as intermediate- or high-intermediate-risk per ESGO–ESTRO–ESP 2021 guidelines who underwent surgery between 2022 and 2023, archival formalin-fixed paraffin-embedded (FFPE) tissue blocks retrieved for POLE mutation testing?
The English language leaves much to be desired in many places and is unclear or approximate (see lines 188-190).
The authors state that: Most patients were diagnosed preoperatively by fractional curettage or endometrial sampling. But what if diagnostic hysteroscopy and avoiding blind diagnosis have been discussed for years (see FIGO guidelines)?
The paragraph from lines 192 to 196 needs to be completely rewritten and clarified.
We need to be precise in staging and percentages, avoiding approximations.
This work seems to me, in its premise, like an excellent restaurant dish,
and then it turns out like a frozen-food soup.
I've thoroughly revised it.
Response
We sincerely thank the reviewer for the careful and thorough evaluation of our manuscript and for recognizing the importance of molecular classification in endometrial cancer. We also appreciate the candid critique, which has helped us substantially improve the clarity, rigor, and presentation of the manuscript. We address each concern point by point below.
- Retrospective design and validity of results
(The study is promising, as it addresses the crucial aspect of molecular classification of endometrial cancer. However, the M&Ms leave much to be desired. First of all, the study is retrospective, which severely limits the validity of the results)
We fully agree that the retrospective nature of this study represents an inherent limitation. This has now been explicitly acknowledged in the revised manuscript.
In the end of introduction part, we added (line 80-84 in the track change version, line 80-85 in the non-track change version)
However, evidence from real-world Asian populations remains limited, particularly in Thailand, where incidence rates are increasing. Therefore, this study aimed to evaluate clinical outcomes and clinicopathologic correlations by molecular subtype among endometrial cancer patients treated at a tertiary care center in Thailand, providing region-specific real-world evidence to support the integration of molecular classification into routine clinical practice
In the Methods section, we added: (headline study design and patient population, line 90-96 in track change version, line 93-99 in no track change version)
This study was designed as a retrospective observational cohort reflecting real-world clinical practice. It was approved by the Institutional Ethics Committee of Chiang Mai University. Medical records of all patients with histologically confirmed endometrial carcinoma (EC) who underwent primary surgical treatment at Chiang Mai University Hospital between January 2015 and December 2023 were retrospectively reviewed. Given the retrospective nature of the study, analyses were exploratory and aimed to describe clinicopathologic characteristics and prognostic patterns rather than to establish causal relationships
In the Limitations section, we further emphasized: (line 498-513 in track- change version, line 417-432 in non- track- change version)
First, the retrospective design inherently limits causal inference and is subject to information and selection bias. Second, molecular testing was not universally performed throughout the study period, reflecting real-world practice during a transitional phase of molecular guideline adoption. This selective testing likely resulted in an underestimation of the true prevalence of POLE-mutated and NSMP tumors.Third, surgical staging practices varied over time, reflecting evolving international guidelines and individualized, risk-adapted decision-making, which may have introduced heterogeneity in clinicopathologic assessment. Fourth, the small number of patients in certain molecular subgroups,particularly POLE-mutated and NSMP tumors,limited statistical power for subgroup analyses and necessitates cautious interpretation of subgroup-specific outcomes.Finally, the relatively short median follow-up duration in the POLE-mutated and NSMP groups may have led to an underestimation of late recurrences.
Accordingly, we tempered our conclusions throughout the manuscript, framing our findings as real-world supportive evidence during a transitional period of molecular adoption, rather than confirmatory evidence comparable to prospective trials.
2. “At discretion” pelvic and/or paraaortic lymphadenectomy
Then, the idea of ​​"at discretion" pelvic and/or lumboaortic lymphadenectomy/sampling doesn't sound right to me. Either you follow the guidelines or you don't.
We appreciate the reviewer’s concern regarding surgical consistency. We added: (line 107-111 in track changed version and line 105-110 in the non-track-change version) in the method part
Standard surgical management consisted of total hysterectomy with bilateral salpingo-oophorectomy and peritoneal washing. Pelvic and/or para-aortic lymph node assessment (sampling or lymphadenectomy) was performed according to tumor risk factors, intraoperative findings, patient comorbidities, and prevailing international guidelines at the time of treatment, reflecting risk-adapted real-world practice over the extended study period.
3. POLE testing limited to intermediate/high–intermediate risk (2022–2023)
But if the study was conducted between January 2015 and December 2023, why were only tumors classified as intermediate- or high-intermediate-risk per ESGO–ESTRO–ESP 2021 guidelines who underwent surgery between 2022 and 2023, archival formalin-fixed paraffin-embedded (FFPE) tissue blocks retrieved for POLE mutation testing?
We appreciate the reviewer’s important question regarding the timing of POLE mutation testing. POLE analysis was limited to cases operated between 2022 and 2023 because adequate DNA quality from archival formalin-fixed paraffin-embedded (FFPE) tissue is essential for reliable POLE exonuclease domain testing. In our institution, older FFPE blocks (particularly those processed before 2022) frequently showed DNA degradation and insufficient yield, precluding accurate molecular analysis. Therefore, to ensure analytical validity and avoid false-negative results, POLE testing was restricted to more recent cases with preserved tissue quality, particularly among patients classified as intermediate- or high–intermediate-risk according to ESGO–ESTRO–ESP 2021 guidelines. This pragmatic approach reflects real-world laboratory constraints and has now been explicitly clarified in the Methods and Limitations sections.
We added this point in the method part (line 123-125 in the track changed version, line 116-120 in the non-track changed version)
……POLE testing was restricted to this period to ensure adequate DNA quality, as older FFPE specimens frequently demonstrated DNA degradation that could compromise the reliability of exonuclease domain sequencing. …..
and added this point in the discussion part (limitation) (line 502-505 in the track changed version and line 421-423 in the non-track changed version)
….In particular, POLE mutation analysis was restricted to more recent surgical cases because adequate DNA quality could not be reliably obtained from older formalin-fixed, paraffin-embedded (FFPE) tissue blocks….
4. English language quality
The English language leaves much to be desired in many places and is unclear or approximate (see lines 188-190).
We acknowledge this concern and apologize for any lack of clarity in the original submission. The manuscript has now undergone comprehensive professional English editing, and multiple sections, including those previously noted (lines 188–190 in the original version),have been rewritten for clarity, precision, and consistency. We believe the revised version substantially improves readability and scientific accuracy.
Former Line 188-190 (present line 204-215)
All statistical analyses were performed using IBM SPSS Statistics version 23.0 (IBM Corp., Armonk, NY, USA). Descriptive statistics were used to summarize patient demographics and clinicopathologic characteristics. Continuous variables are presented as mean ± standard deviation, and categorical variables as frequencies and percentages. Comparisons between categorical variables were conducted using the chi-square test or Fisher’s exact test, as appropriate. Variables demonstrating a p value < 0.05 in univariate analysis were subsequently entered into multivariable binary logistic regression models to identify independent factors associated with MMR deficiency and p53 abnormality. Survival outcomes were estimated using the Kaplan–Meier method and compared across molecular subtypes using the log-rank test. All statistical tests were two-sided, and a p value < 0.05 was considered statistically significant.
5. Preoperative diagnosis: curettage vs. hysteroscopy
The authors state that: Most patients were diagnosed preoperatively by fractional curettage or endometrial sampling. But what if diagnostic hysteroscopy and avoiding blind diagnosis have been discussed for years (see FIGO guidelines)?
We thank the reviewer for this important comment. We fully agree that diagnostic hysteroscopy with directed biopsy is recommended by FIGO and other international guidelines to improve diagnostic accuracy and to avoid blind sampling when feasible.
However, during the study period (2015–2023), preoperative diagnostic approaches in our institution reflected real-world clinical practice, in which fractional curettage or endometrial sampling (including pipelle biopsy) remained the most commonly used diagnostic methods, particularly in patients presenting with abnormal uterine bleeding. Diagnostic hysteroscopy was selectively performed based on clinical indication, availability of equipment, patient condition, and operator expertise, and was not uniformly applied to all patients.
To address this concern and improve clarity, we have revised the Methods section to explicitly state that preoperative diagnosis was established using fractional curettage, office endometrial sampling, with or without hysteroscopy, depending on clinical circumstances. We have also clarified that the choice of diagnostic modality reflected institutional practice patterns during the study period rather than deviation from guideline recommendations.
Line 101-104 in the track changed version and line 100-103 in the non-track changed version
Most patients were diagnosed preoperatively by fractional curettage or endometrial sampling, with diagnostic hysteroscopy performed selectively based on clinical indication, resource availability, and operator expertise during the study period. Final histopathologic classification and FIGO staging were determined from definitive hysterectomy specimens.
Importantly, all final diagnoses, staging, and molecular analyses in this study were based on definitive hysterectomy specimens, which mitigates potential limitations related to the initial diagnostic method.
6. Paragraph lines 192–196: staging and percentages
The paragraph from lines 192 to 196 needs to be completely rewritten and clarified.
We need to be precise in staging and percentages, avoiding approximations.
Thank you for this important comment. We agree that the original wording was imprecise. We have fully rewritten the Results paragraph describing baseline clinicopathologic characteristics to present exact numbers and percentages, with explicit FIGO 2018 staging categories and clearly defined clinicopathologic variables. All data are now reported precisely and are cross-referenced to Supplementary Table S1, ensuring transparency and reproducibility. The revised text replaces the original lines 192–196 in full.
Result part: Line 239-264 in the track changed version and line 218-243 in the non-track changed version
Between January 2015 and December 2023, a total of 803 patients were diagnosed with endometrial carcinoma at Chiang Mai University Hospital; 184 patients met the inclusion criteria and were included in the final analysis (Table S1). The mean age was 61.3 ± 9.7 years, and the mean body mass index (BMI) was 24.5 ± 5.1 kg/m². The most common presenting symptom was postmenopausal bleeding (82.6%).
Comorbidities were present in 70.1% of patients, most frequently diabetes mellitus, hypertension, and dyslipidemia, and 15.2% had a history of another malignancy, most commonly breast cancer. Preoperative tissue diagnosis was obtained by fractional curettage or endometrial sampling in 92.9% of cases.
Most patients underwent exploratory laparotomy (89.7%), while 10.3% were managed laparoscopically. Pelvic lymph node assessment was performed in 75.0%, and paraaortic lymph node assessment in 26.6%. Complete macroscopic resection was achieved in 91.8% of patients.
According to FIGO 2018 staging, disease stages were distributed as follows: IA 32.6%, IB 25.5%, II 6.0%, IIIA 8.7%, IIIB 3.8%, IIIC1 8.7%, IIIC2 5.4%, and IVB 9.2%. Endometrioid carcinoma was the predominant histologic subtype (57.1%), followed by high-grade serous carcinoma (30.4%), clear cell carcinoma (6.5%), and carcinosarcoma (6.0%). High-grade tumors (grade 3) accounted for 63.0% of cases.
Myometrial invasion ≥50% was observed in 38.6%, lymphovascular space invasion (LVSI) was present in 59.2%, and endocervical involvement occurred in 23.4%. Pelvic and paraaortic lymph node metastases were identified in 14.7% and 3.8%, respectively. Based on ESGO–ESTRO–ESP 2021 risk classification, 57.6% of patients were categorized as high risk.
Neoadjuvant chemotherapy was administered in 7.6%, and 79.3% received adjuvant treatment following surgery. During follow-up, 15.8% experienced disease recurrence, and 28.3% died.
Final remark
This work seems to me, in its premise, like an excellent restaurant dish,and then it turns out like a frozen-food soup.I've thoroughly revised it.
We appreciate the reviewer’s candid and thoughtful feedback and understand the concern conveyed by this metaphor. We fully agree that, while the scientific premise of the study is strong, the original presentation did not consistently reflect this potential with sufficient clarity and rigor.
In response, we have extensively revised the manuscript to improve methodological transparency, precision of reporting, and coherence across sections. Specifically, we have:
- Clarified the retrospective, real-world design and explicitly framed the study as exploratory rather than confirmatory.
- Rewritten the Methods and Results sections to ensure precise definitions, exact numbers and percentages, and alignment with current guidelines.
- Strengthened the Discussion by systematically comparing our findings with major Western and international datasets and by tempering conclusions where subgroup sizes are limited.
- Carefully edited the manuscript for language, structure, and clarity, including the sections specifically highlighted by the reviewer.
We believe these revisions substantially improve the consistency between the study’s premise and its execution and presentation. We are grateful for the reviewer’s thorough evaluation, which has helped us significantly enhance the quality and clarity of the manuscript.

Reviewer 3 Report
Comments and Suggestions for Authors
In this manuscript Salakphet et al, have assessed 184 Thai patients suffering from endometrial cancer. They have proposed the use of molecular testing to predict prognosis. This is an important study, however there is a major limitation that needs to be addressed.
1) The authors need to show the immunohistochemistry data of various patients that was used to conclude the findings.
Author Response
Comments and Suggestions for Authors
Reviewer 3
Comments and Suggestions for Authors
In this manuscript Salakphet et al, have assessed 184 Thai patients suffering from endometrial cancer. They have proposed the use of molecular testing to predict prognosis. This is an important study, however there is a major limitation that needs to be addressed.
- The authors need to show the immunohistochemistry data of various patients that was used to conclude the findings.
Response:
We thank the reviewer for this valuable comment and agree that clear presentation of immunohistochemistry (IHC) data is essential for interpretation of the molecular classification.
In response, we have revised Table 1 to more explicitly present the molecular testing results underlying subtype assignment. Specifically, we have revised the table title to emphasize that it summarizes IHC results (MMR and p53) together with POLE testing, and we have added a detailed footnote clarifying how MMR status, p53 expression patterns, and POLE mutation results were integrated using a predefined hierarchical algorithm to determine the final molecular subtype. We have also clarified that all IHC slides were independently reviewed by two gynecologic pathologists.
In addition, we have revised the Results text to explicitly direct readers to Table 1 as the source of the IHC and POLE data used for molecular classification.
Line 285-293 in the track changed version and line 245-256 in non-track changed version
The immunohistochemistry (IHC) results for mismatch repair (MMR) proteins and p53, together with POLE mutation testing, which collectively formed the basis of final molecular subtype assignment, are summarized in Table 1.

Round 2
Reviewer 1 Report
Comments and Suggestions for Authors
Thank you for addressing my comments.